# Comparative analysis of oral health behaviour and utilisation of oral health care services in the general population and among patients with non-communicable diseases in Korea: a repeated cross-sectional survey conducted from 2008 to 2022

Jeehee Pyo[1,2☯], Hyeran Jeong[1☯], Noor Afif Mahmudah[1], Young-Kwon Park[3], Minsu Ock [1,3,4]*

1 Department of Preventive Medicine, University of Ulsan College of Medicine, Seoul, Republic of Korea, 2 Always be with you, The PLOCC Affiliated Counseling Training Center, Seoul, Republic of Korea, 3 Prevention and Management Center, Ulsan Regional Cardiocerebrovascular Center, Ulsan University Hospital, Ulsan, Republic of Korea, 4 Department of Preventive Medicine, Ulsan University Hospital, University of Ulsan College of Medicine, Ulsan, Republic of Korea

☯ These authors contributed equally to this work.
* ohohoms@naver.com

## Abstract

### Background

Understanding the oral health behaviour and utilisation of oral health care services among patients with non-communicable diseases (NCDs) is essential for the development of oral health care management services for these patients. In this study, we comparatively analysed the trends in oral health behaviour and oral health care service utilisation among patients with various NCDs and the general population.

### Methods

We analysed data obtained via the Korea Community Health Survey from 2008 to 2022. Comparative analyses of the general population and patients with 15 different NCDs, including diabetes mellitus and depression, were conducted for the following variables: toothbrushing practice, use of dental floss and interdental brushes, annual scaling (tartar removal), and annual oral examinations. Joinpoint regression analyses were used to assess for statistically significant changes in oral health behaviour and oral health care service utilisation according to year and region.

### Results

Overall, oral health behaviour steadily improved in the general population and among patients with NCDs. However, the rates of toothbrushing before going to bed in

**Data availability statement:** All data generated or analyzed during this study are included in this published article and its Supporting Information files.

**Funding:** This study was supported by a grant from the National R&D Program for Cancer Control, Ministry of Health & Welfare, Republic of Korea (HA21C0107). The funders had no role in study design, data collection and analysis, decision to publish, or preparation of the manuscript.

**Competing interests:** The authors declare that they have no competing interests.

patients with hypertension (90.0%) and diabetes mellitus (88.7%) were still lower than that in the general population (92.9%) in 2022. Regarding oral health service utilisation, the rates of annual scaling and oral examination among patients with NCDs, apart from those with dyslipidaemia, were lower than those in the general population. For example, in 2017, the rates of annual scaling and oral examination of patients with diabetes mellitus were 43.4% and 36.5%, respectively. These rates were lower than those in the general population, at 47.2% and 43.0%, respectively. In terms of inter-regional variations in oral health-related indicators, considerable inter-regional variations were observed in the oral health behaviour and oral health care service utilisation of patients with NCDs.

## Conclusion

This study highlighted that the practice and utilisation rates of oral health behaviour and oral health care services, respectively, among patients with NCDs have increased. However, in comparison with those of the general population, further improvements are necessary. A practical solution could be to establish a system that provides all necessary oral-related services, such as patient education and oral examinations, to NCDs patients through medical-dental integration or oral medical care coordination.

## Introduction

Oral health is an essential component of overall health [1]. Oral disease is a non-communicable disease (NCD) that causes pain and discomfort, resulting in a negative impact on daily life. Moreover, oral disease causes or worsens other NCDs [2,3]. Twenty-eight NCDs, including type 2 diabetes mellitus, depression, asthma, and various types of cancer, are strongly associated with oral diseases [3]. This is because smoking, drinking, and obesity, which are representative risk factors for NCDs, are also risk factors for oral diseases [4–6], and because bacteraemia and systemic inflammation caused by oral diseases affect the entire body [7]. Additionally, the deterioration of oral health affects dietary patterns, resulting in nutritional imbalances, which have negative impacts on overall health [8]. Therefore, providing education throughout the community regarding appropriate oral health behaviour is fundamental.

Furthermore, improving access to oral health care services reduces the burden of oral diseases and major NCDs [1,9]. Toothbrushing, a cornerstone of oral health behaviour, effectively prevents dental caries and periodontal disease [10]. Dental plaque is frequently difficult to completely remove via toothbrushing alone. Thus, interdental cleaning using dental floss and interdental brushes is recommended, to achieve more thorough plaque removal [11]. Regular dental visits are important, as preventive activities, which include oral examinations and scaling (tartar removal), can prevent oral diseases such as dental caries and periodontitis [12,13].

In order to promote desirable oral health behaviours and improve accessibility to oral health care services, we need to understand the current oral health status of various population groups, including infants, young children, and older adults [14]. In Korea, numerous health-related surveys are conducted at the national level, which include questions regarding oral health behaviour and unmet needs in terms of oral health care services [15,16]. These surveys are used to analyse various aspects of oral health of different population groups, including adolescents, older adults, and patients with cancer [17–19]. However, few comprehensive examinations of the oral health behaviours and oral health care service utilisation of patients with chronic diseases have been published.

Based on the two-way relationship between oral diseases and NCDs, such studies among patients with chronic diseases are essential. In addition, in comparison with the general population, such patients tend to exhibit unhealthy behaviours, such as smoking, drinking, and consuming large amounts of salt [20,21]. Therefore, in this study, we aimed to compare the oral health behaviour and oral health care service utilisation among patients with different NCDs and the general population.

## Materials and methods

### Study design

We analysed data from the Korea Community Health Survey (KCHS) conducted from 2008 to 2022. Under management of the Korea Disease Control and Prevention Agency, through the KCHS, local government affiliated public health centres collect health-related and health behaviour data, including data regarding smoking and drinking, NCDs, and medication use [22,23]. The survey targeted individuals older than 19 years. Approximately 900 individuals per year are surveyed at each public health centre, for an annual total of > 200 000 individuals.

### Classification of NCDs and defining the general population

The 15 NCDs included in this study were as follows: hypertension, diabetes mellitus, dyslipidaemia, stroke, angina, myocardial infarction, a combination of angina and myocardial infarction, arthritis, osteoporosis, asthma, allergic rhinitis, atopic dermatitis, cataracts, depression, and cancer. Patients were defined as having an NCD if they answered 'yes' to the question regarding whether they had ever been diagnosed with a particular NCD by a doctor. However, patients who answered 'yes' to whether they had depression were categorised into those with either situational or clinical depression, as well as those who had been diagnosed by a doctor. The NCDs differed based on the year that the KCHS was conducted (Supplementary File 1). "The general population" was operationally defined as all those who had participated in the KCHS and did not have NCDs.

### Oral health behaviour and oral health care service utilisation

The methodology regarding asking about toothbrushing differed depending on the year in which the KCHS was conducted (Supplementary File 2). For example, the timing of toothbrushing practice encompassed the following responses: 'After breakfast, yesterday'; 'After lunch, yesterday'; 'After dinner, yesterday'; 'Before going to bed yesterday'; or a combination of these. The response, 'after lunch, yesterday' was available each year from 2008 to 2022. However, the response, 'before going to bed, yesterday' was only available each year from 2008 to 2009, 2014–2019, and thereafter in 2021. The options available as responses to the question regarding toothbrushing were 'yes', 'no', 'did not eat', and 'did not sleep'. The dichotomous options available as responses in 2008, 2011, and 2013, were 'yes' and 'no' to the question, 'In addition to toothpaste and toothbrushes, do you usually use dental floss or interdental brushes, to keep your teeth healthy?'

Regarding oral health care service utilisation, responses up to the year 2017 were collected. The dichotomous responses were 'yes' and 'no' to the question, 'Have you had scaling or tartar removal in the past year?' Moreover, the dichotomous responses were 'yes' and 'no' to the question, 'No special problems have been observed with your mouth in the past year; however, have you ever undergone an oral examination, to assess your oral health?'

## Data analysis

By using descriptive statistics, the oral health behaviour and oral health care service utilisation were evaluated according to year and region, at the city and provincial levels. Moreover, different rates were compared between patients with NCDs and the general population. Joinpoint regression analyses were used to evaluate changes in oral health behaviour and oral health care service utilisation according to year and region [24]. The number of joinpoints was set to ≤ 1; however, we frequently referred to the numbers recommended by the statistical analysis program. Therefore, the annual percentage change (APC) and average APC (AAPC) when the number of joinpoints was > 1 were calculated, and p-values were presented.

Data were analysed using Microsoft Excel 2016 (Microsoft Corporation, Seattle, WA, USA). Descriptive statistical analyses were performed using Stata/SE13.1 (StataCorp, Texas, TX, USA). Joinpoint regression analyses were performed using the Joinpoint Regression Program, version 5.0.2 (US National Cancer Institute, Bethesda, MD, USA).

## Ethical considerations

This study was conducted without the approval from an institutional review board, and informed consent was not obtained, because we used publicly available and anonymised data from the Korea Community Health Survey (KCHS).

## Results patient characteristics

The annual distributions of the general population and patients with NCDs are presented in Table 1. Among the 15 NCDs, the disease categories identified in all observation years were hypertension, diabetes mellitus, and depression (symptom experience). As of 2022, 70,219 patients with hypertension and 30,784 patients with diabetes mellitus were included in the analysis.

## Oral health behaviour: toothbrushing practice

Overall, oral health behaviour steadily improved in the general population and among patients with NCDs. However, in most cases, the oral health behaviour of patients with NCDs was worse than that of the general population. Regarding toothbrushing after lunch, the practice rate for the general population increased by 19.8p% (29.4%), between 2008 and 2022 (Fig 1). Moreover, regarding toothbrushing after lunch for patients with hypertension and those with diabetes mellitus, the practice rates increased by 26.2p% (43.5%) and 25.2p% (43.9%), respectively, over the same period. However, the practice rate of toothbrushing after lunch among patients with NCDs remained lower than that in the general population.

Regarding toothbrushing before going to bed, the practice rate for the general population increased by 54.6p% (58.7%), between 2008 and 2022 (Fig 2). Furthermore, regarding toothbrushing before going to bed among patients with hypertension and diabetes mellitus, the practice rates increased by 59.0p% (65.6%) and 59.0p% (66.5%), respectively, over the same period. Statistically significant improvements in toothbrushing before going to bed were noted in both the general population and patients with NCDs in 2022, compared with 2008. However, the practice rate of toothbrushing before going to bed remained higher in the general population than that among patients with NCDs.

Regarding the annual trends, the practice rate of toothbrushing after lunch in the general population increased until 2011 (APC = 10.58, p = 0.069) and decreased from 2011 until 2016 (APC = -3.27, p = 0.300). However, from 2016 to 2022, an increasing trend was observed (APC = 4.54, p = 0.026). A similar pattern was observed among patients with NCDs from 2016 to 2022, such as among those with hypertension (AAPC = 4.57, p = 0.043) and those with diabetes mellitus (AAPC = 4.83, p = 0.060). The practice rate of toothbrushing before going to bed increased annually in the general population (AAPC = 5.84, p < 0.001), patients with hypertension (AAPC = 7.40, p < 0.001), and those with diabetes mellitus (AAPC = 7.53, p < 0.001). Further details are presented in Supplementary File 3.

**Table 1. The number of patients by year by disease category.**

| Disease | Number of patients by year (%) | | | | | | | | | | | | | | | Total |
|---|---|---|---|---|---|---|---|---|---|---|---|---|---|---|---|---|
| | 2008 | 2009 | 2010 | 2011 | 2012 | 2013 | 2014 | 2015 | 2016 | 2017 | 2018 | 2019 | 2020 | 2021 | 2022 | |
| Hypertension | 42,102 (4.9) | 45,258 (5.3) | 48,600 (5.6) | 51,581 (6.0) | 52,710 (6.1) | 55,304 (6.4) | 55,727 (6.5) | 58,045 (6.7) | 59,263 (6.9) | 62,115 (7.2) | 65,055 (7.6) | 65,308 (7.6) | 63,695 (7.4) | 65,907 (7.7) | 70,219 (8.2) | 860,889 (100.0) |
| Diabetes mellitus | 15,290 (4.5) | 16,520 (4.8) | 18,255 (5.3) | 19,276 (5.6) | 19,880 (5.8) | 20,848 (6.1) | 21,846 (6.4) | 22,852 (6.7) | 23,772 (7.0) | 25,115 (7.4) | 25,321 (7.4) | 26,408 (7.7) | 26,717 (7.8) | 28,355 (8.3) | 30,784 (9.0) | 341,239 (100.0) |
| Dyslipidaemia | – | – | – | 21,939 (10.6) | 24,361 (11.8) | 25,519 (12.3) | 28,885 (13.9) | 31,652 (15.3) | 35,333 (17.0) | 39,552 (19.1) | – | – | – | – | – | 207,241 (100.0) |
| Stroke | 4,189 (11.8) | 4,558 (12.8) | 4,397 (12.4) | 4,389 (12.3) | 4,290 (12.1) | 4,399 (12.4) | 4,455 (12.5) | – | 4,916 (13.8) | – | – | – | – | – | – | 35,593 (100.0) |
| MI | 2,473 (14.6) | 2,539 (15.0) | 2,822 (16.6) | 2,993 (17.6) | 3,170 (18.7) | 2,977 (17.5) | – | – | – | – | – | – | – | – | – | 16,974 (100.0) |
| Angina pectoris | 3,088 (13.7) | 3,338 (14.9) | 3,871 (17.2) | 3,938 (17.5) | 4,131 (18.4) | 4,108 (18.3) | – | – | – | – | – | – | – | – | – | 22,474 (100.0) |
| MI and angina pectoris | – | – | – | – | – | – | 6,794 (47.9) | – | 7,387 (52.1) | – | – | – | – | – | – | 14,181 (100.0) |
| Arthritis | 24,578 (8.5) | 26,273 (9.1) | 27,846 (9.7) | 28,814 (10.0) | 28,463 (9.9) | 29,685 (10.3) | 28,733 (10.0) | 30,386 (10.6) | 29,173 (10.1) | 33,932 (11.8) | – | – | – | – | – | 287,883 (100.0) |
| Osteoporosis | 11,841 (10.0) | 14,658 (12.4) | 16,561 (14.0) | 18,072 (15.3) | 18,320 (15.5) | 19,221 (16.3) | – | – | 19,432 (16.5) | – | – | – | – | – | – | 118,105 (100.0) |
| Asthma | 5,346 (12.7) | 5,344 (12.7) | 5,989 (14.2) | 6,148 (14.6) | 6,349 (15.1) | 6,372 (15.1) | – | 6,618 (15.7) | – | – | – | – | – | – | – | 42,166 (100.0) |
| Allergic rhinitis | 9,464 (6.2) | 20,100 (13.2) | 21,718 (14.3) | 23,350 (15.4) | 24,398 (16.1) | 25,776 (17.0) | – | 27,008 (17.8) | – | – | – | – | – | – | – | 151,814 (100.0) |
| Atopic dermatitis | 2,480 (7.3) | 4,928 (14.4) | 5,214 (15.3) | 5,208 (15.2) | 5,339 (15.6) | 5,350 (15.7) | – | 5,646 (16.5) | – | – | – | – | – | – | – | 34,165 (100.0) |
| Cataracts | 14,007 (9.5) | 16,282 (11.0) | 18,555 (12.6) | 22,648 (15.3) | 22,394 (15.2) | 24,119 (16.3) | – | – | – | 29,788 (20.2) | – | – | – | – | – | 147,793 (100.0) |
| Depression (physician diagnosis) | – | 6,028 (20.9) | 5,306 (18.4) | 5,805 (20.1) | 5,620 (19.5) | 6,120 (21.2) | – | – | – | – | – | – | – | – | – | 28,879 (100.0) |
| Depression (symptom experience) | 19,245 (8.8) | 17,411 (7.9) | 12,732 (5.8) | 11,832 (5.4) | 11,793 (5.4) | 13,377 (6.1) | 16,129 (7.3) | 14,968 (6.8) | 14,012 (6.4) | 14,248 (6.5) | 13,268 (6.0) | 14,071 (6.4) | 12,839 (5.8) | 16,332 (7.4) | 17,558 (8.0) | 219,815 (100.0) |

*(Continued)*

**Table 1.** (Continued)

| Disease | Number of patients by year (%) | | | | | | | | | | | | | | |
| --- | --- | --- | --- | --- | --- | --- | --- | --- | --- | --- | --- | --- | --- | --- | --- |
| | 2008 | 2009 | 2010 | 2011 | 2012 | 2013 | 2014 | 2015 | 2016 | 2017 | 2018 | 2019 | 2020 | 2021 | 2022 | Total |
| Prevalence of depressive symptoms (PHQ-9) | – | – | – | – | – | – | – | – | – | 7,705 (16.3) | 8,648 (18.3) | 7,477 (15.9) | 6,436 (13.6) | 7,745 (16.4) | 9,144 (19.4) | 47,155 (100.0) |
| Cancer | 4,228 (35.3) | – | – | – | 7,759 (64.7) | – | – | – | – | – | – | – | – | – | – | 11,987 (100.0) |
| General population | 220,258 (6.7) | 230,715 (6.7) | 229,229 (6.7) | 229,226 (6.7) | 228,921 (6.7) | 228,781 (6.7) | 228,712 (6.7) | 228,558 (6.7) | 228,452 (6.7) | 228,381 (6.7) | 228,340 (6.7) | 229,099 (6.7) | 229,269 (6.7) | 229,242 (6.7) | 231,785 (6.8) | 3,428,968 (100.0) |

Abbreviations: MI: myocardial infarction; PHQ-9: Patient Health Questionnaire-9

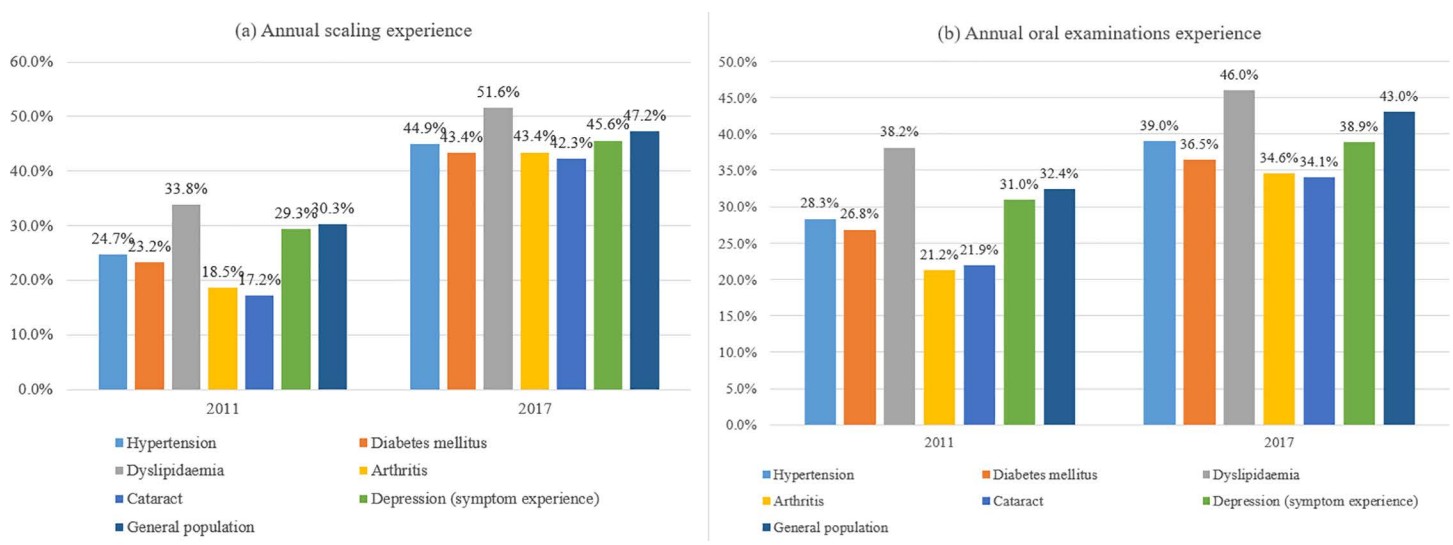

**Fig 1. Toothbrushing practice after lunch among the general population and patients with NCDs.**

Abbreviations: NCDs = non-communicable diseases

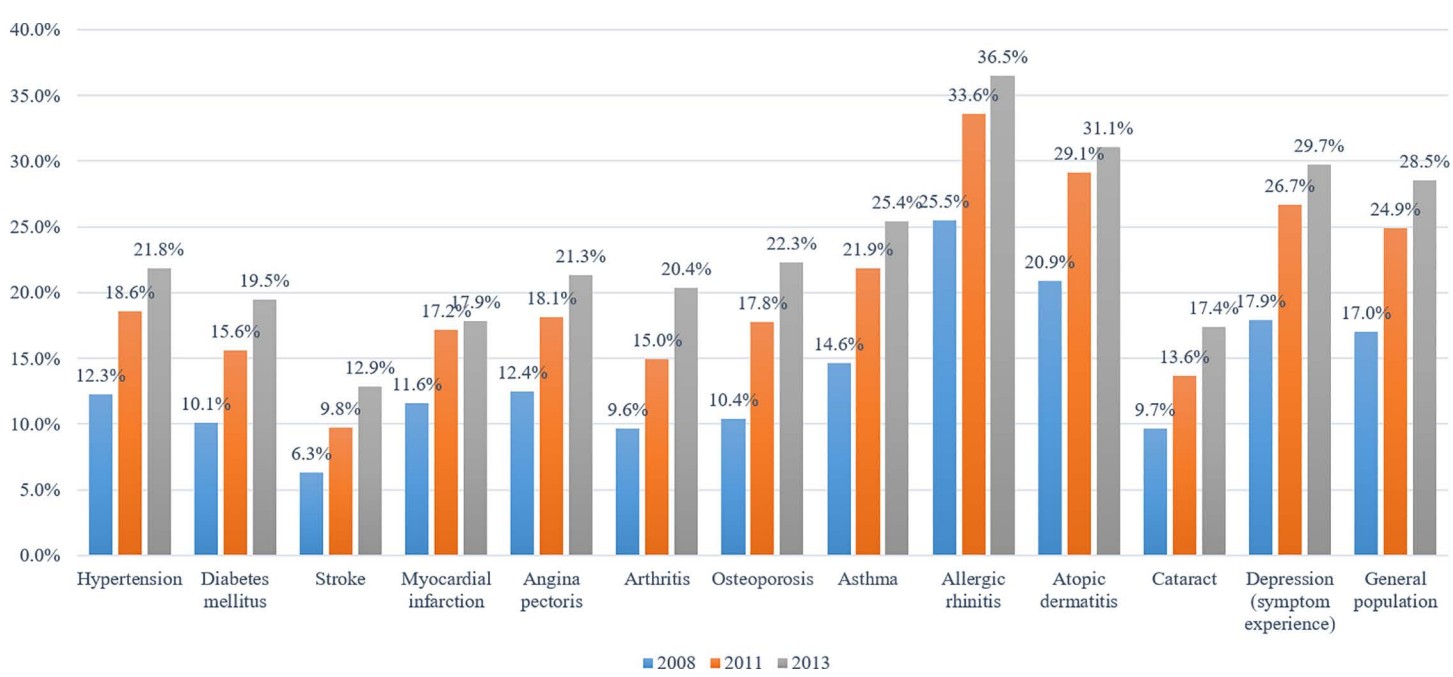

**Fig 2. Toothbrushing practice before going to bed among the general population and patients with NCDs.**

Abbreviations: NCDs = non-communicable diseases

### Oral health behaviour: use of dental floss and interdental brushes

Regarding dental floss and interdental brushes, the usage rate among patients with NCDs, apart from those with allergic rhinitis, atopic dermatitis, and depression, remained lower than that in the general population throughout the study period (Fig 3). The most recent data regarding dental floss and interdental brushes were obtained in 2013, with usage rates of

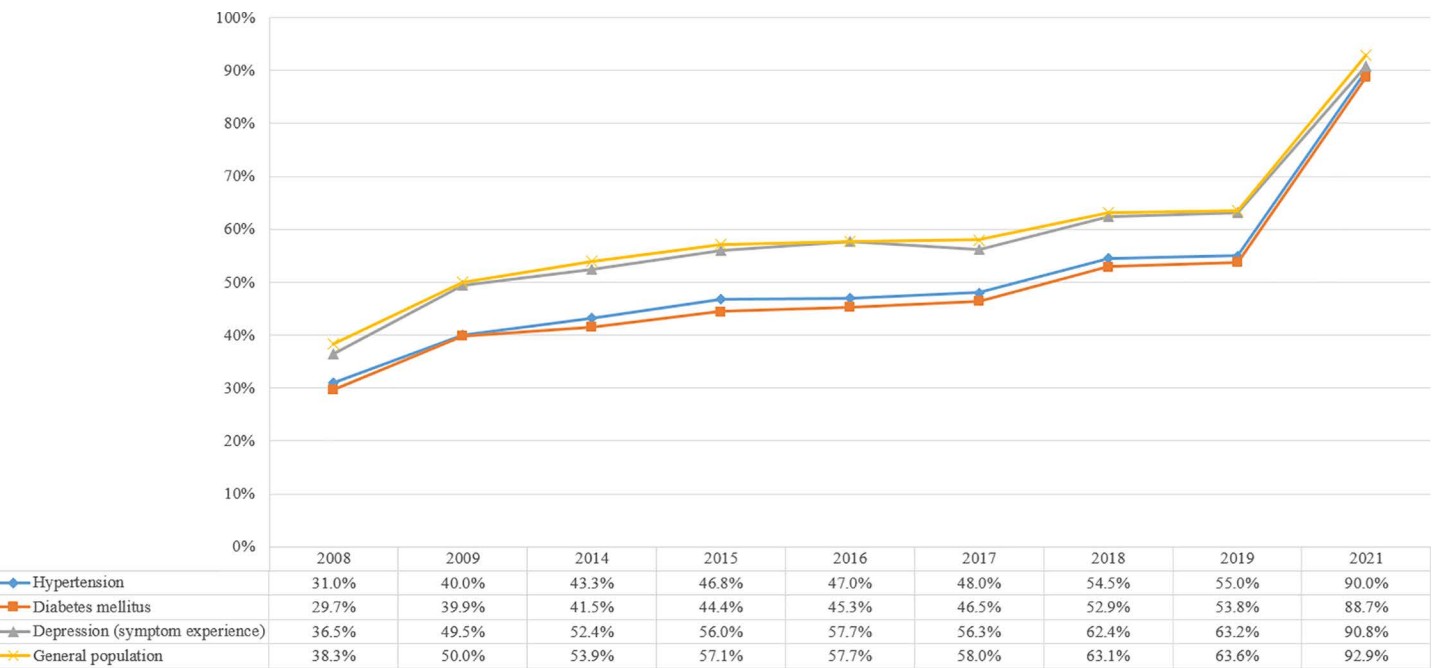

**Fig 3. Dental floss and interdental brush usage rates among the general population and patients with NCDs.**

Abbreviations: NCDs = non-communicable diseases

21.8% and 19.5%, respectively, among patients with hypertension and diabetes mellitus. Overall, an increasing trend in the usage rates of dental floss and interdental brushes was observed in the general population and among patients with NCDs, from 2008 to 2013. In particular, a statistically significant increase in the usage rate was observed among patients with diabetes mellitus (APC = 14.18, *p* = 0.047). Further details are presented in Supplementary File 4.

### Oral health service utilisation: annual scaling and oral examinations

Similar results were revealed regarding annual scaling and oral examinations (Fig 4). The utilisation rates of annual scaling and oral examinations among patients with NCDs, apart from those with dyslipidaemia, were lower than those in the general population. In 2017, the utilisation rates of annual scaling and oral examinations among patients with diabetes mellitus were 43.4% and 36.5%, respectively. A clear improvement was observed compared to those in 2011, with utilisation rates of 23.2% and 26.8%, respectively. Nonetheless, in 2017, the utilisation rates of annual scaling and oral examinations among patients with diabetes mellitus were lower than those of the general population, which were 47.2% and 43.0%, respectively.

The utilisation rate of annual scaling steadily increased from 2008 to 2017. Statistically significant changes were observed in the general population (APC = 8.39, *p* < 0.001), among patients with hypertension (APC = 10.72, *p* < 0.001), and among those with diabetes mellitus (APC = 11.38, *p* < 0.001). Additionally, the utilisation rate of annual oral examinations steadily increased from 2008 to 2017. In particular, a rapid increase in this rate was observed from 2008 to 2010. Thereafter, a decrease was observed; however, overall trend was upward. Overall, statistically significant increases in the utilisation rates of annual oral examinations were observed in the general population (AAPC = 9.90, *p* < 0.001), among patients with hypertension (AAPC = 10.42, *p* < 0.001), and among those with diabetes mellitus (AAPC = 10.33, *p* < 0.001). Further details are presented in Supplementary File 5.

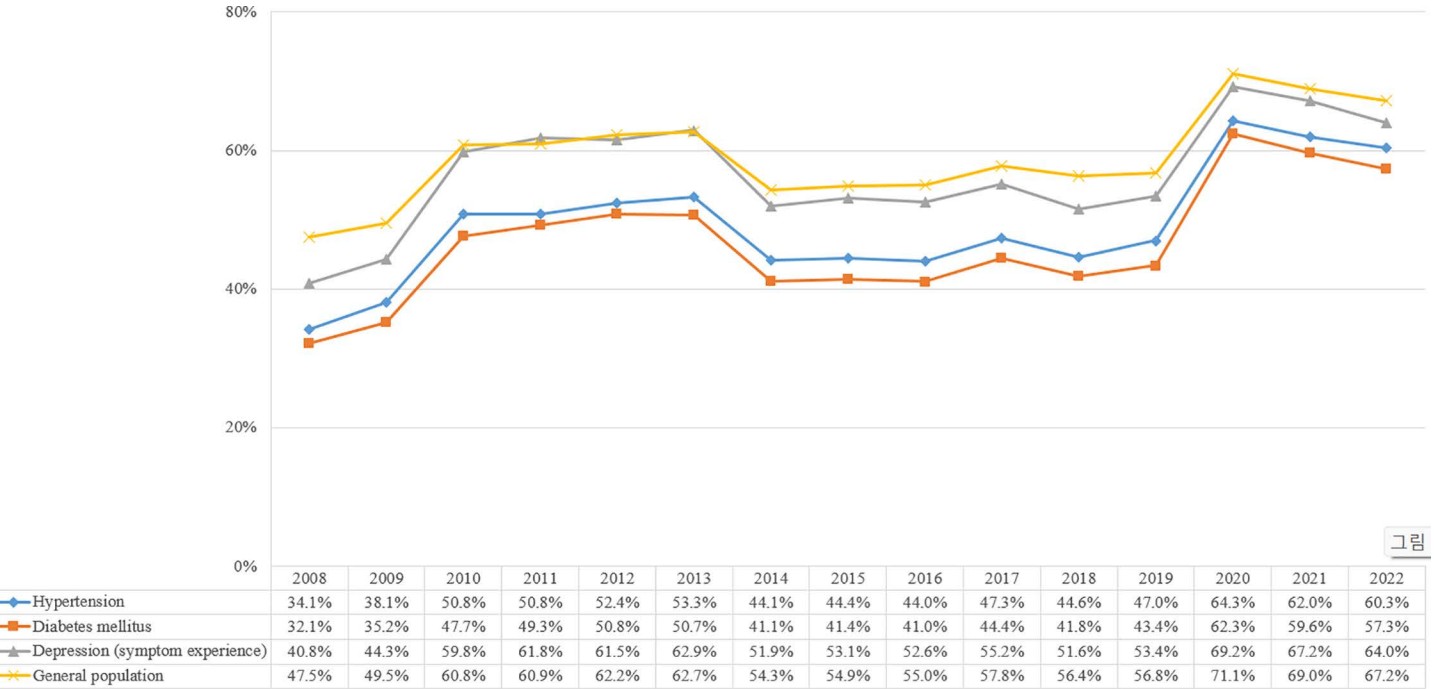

**Fig 4. Rates of oral health service utilisation among the general population and patients with NCD** s: **(a) annual scaling experience and (b) annual oral examinations experience.**

Abbreviations: NCDs = non-communicable diseases

| | 2008 | 2009 | 2010 | 2011 | 2012 | 2013 | 2014 | 2015 | 2016 | 2017 | 2018 | 2019 | 2020 | 2021 | 2022 |
|---|---|---|---|---|---|---|---|---|---|---|---|---|---|---|---|
| Hypertension | 34.1% | 38.1% | 50.8% | 50.8% | 52.4% | 53.3% | 44.1% | 44.4% | 44.0% | 47.3% | 44.6% | 47.0% | 64.3% | 62.0% | 60.3% |
| Diabetes mellitus | 32.1% | 35.2% | 47.7% | 49.3% | 50.8% | 50.7% | 41.1% | 41.4% | 41.0% | 44.4% | 41.8% | 43.4% | 62.3% | 59.6% | 57.3% |
| Depression (symptom experience) | 40.8% | 44.3% | 59.8% | 61.8% | 61.5% | 62.9% | 51.9% | 53.1% | 52.6% | 55.2% | 51.6% | 53.4% | 69.2% | 67.2% | 64.0% |
| General population | 47.5% | 49.5% | 60.8% | 60.9% | 62.2% | 62.7% | 54.3% | 54.9% | 55.0% | 57.8% | 56.4% | 56.8% | 71.1% | 69.0% | 67.2% |

## Inter-regional variations in oral health-related indicators

Based on the latest oral health-related indicators, regional disparities tended to be greater among patients with NCDs than among individuals in the general population, apart from the practice rate of toothbrushing after breakfast (Supplementary File 3). In 2019, the province of Jeollabuk-do had the highest practice rate of toothbrushing after dinner, at 79.8%, in the general population. Conversely, the city of Daegu had the lowest rate, at 54.2%. Moreover, the cities of Daejeon and Daegu had the highest and lowest practice rates of toothbrushing after dinner among patients with hypertension, respectively, at 79.4% and 66.0%; and among those with diabetes mellitus, at 80.6% and 63.2%, respectively.

In particular, regional differences in the utilisation rates of annual oral examinations were substantial (Supplementary File 5). In 2017, regarding the general population, the utilisation rates of annual oral examinations were the highest and lowest in Sejong City and the province of Jeollanam-do, at 56.2% and 30.8%, respectively, with a regional gap of 25.4%. Furthermore, the utilisation rates of annual oral examinations for patients with arthritis, cataracts, and hypertension were the highest in Sejong City, at 56.3%, 48.1%, and 58.6%, respectively; and the lowest in the province of Gyeongsangbuk-do, at 19.0%, 15.8%, and 24.2%, respectively. Additionally, among patients with arthritis, cataracts, and hypertension, with utilisation rates of annual oral examinations at 37.2%, 32.3%, and 34.4%, respectively, large regional gaps were observed.

## Discussion

In this study, we aimed to compare these items between patients with NCDs and the general population. Overall, oral health behaviour tended to improve over the study period. However, by and large, the oral health behaviour of patients with NCDs was poor compared with that of the general population. The utilisation rate of oral health care services

improved; nonetheless, this was predominantly lower among patients with NCDs than that in the general population. Furthermore, considerable inter-regional variations were observed in the oral health behaviour practices and oral health care service utilisation of patients with NCDs, signifying the urgent need for corrective measures.

To date, most studies in the field have been focused on age-related population groups [17,18] rather than on patients with chronic diseases. Some studies have been conducted on patients with specific chronic diseases, such as cancer [19], and others have focused on the relationship between chronic diseases and oral health behaviour. This study is of great significance, because we are not aware of other studies in which the oral health behaviour and trends in oral health service utilisation of patients with different types of NCDs have been examined. Furthermore, the results of this study may be generalizable to all Koreans, as we analysed 15 years of data from the KCHS, which includes > 200 000 participants per year. This methodological rigor lends validity to the study.

## Interpretation and implications of principal findings

In this 15-year trend analysis, we were encouraged by the improving rates of toothbrushing practice and usage of dental floss and interdental brushes in Korea. The practice rate of toothbrushing after lunch, the data of which was available for all 15 years, statistically significantly increased from 2008 to 2022, among patients with diabetes mellitus and hypertension, respectively. However, compared with the practice rate of the general population, the need for further improvement is evident. Toothbrushing is known to reduce the risk of developing hypertension and diabetes mellitus [25]. Consequently, further education is required to improve the oral health behaviour among patients with NCDs, to prevent the development of additional comorbidities.

The most recent data regarding the usage rates of dental floss and interdental brushes were obtained in 2013. Nonetheless, these rates also appear to be improving in the Korean population. However, the absolute figures and the percentage change remained lower for most patients with NCDs than that in the general population, and further education involving the usage of dental floss and interdental brushes among patients with NCDs is necessary. Notably, in 2013, the usage rate of dental floss and interdental brushes among patients with stroke was only 12.9%. Although a seemingly upward trend was observed from 2008 to 2013, achieving a usage rate that exceeds 30% may be challenging. The use of dental floss and interdental brushes is generally recommended for interdental hygiene [11]. Nevertheless, the usage rate of these tools remains low in Korea, and an improvement is urgently needed [26]. If a patient with stroke has difficulty using dental floss and interdental brushes correctly, education regarding their appropriate use should be provided for both the patient and their caregivers [27].

Consistent with the findings of oral health behaviour, the utilisation rate of annual scaling and oral examinations increased during the study period. However, except for those among patients with dyslipidaemia, the utilisation rates of annual scaling and oral examinations were lower among patients with NCDs than those in the general population. Currently, oral examinations are provided at no cost to the individual in Korea. In addition, scaling is covered by health insurance once a year, with a co-payment rate of 30%. These policies have assisted in improving recent utilisation rates of scaling and oral examinations [28,29]. Nevertheless, more guidance and education regarding scaling and the need for oral examinations is required for patients with NCDs.

Overall, the oral health behaviour of patients with NCDs was less satisfactory than that of the general population in this study, and the utilisation rate of oral health care services was lower. One reason for these results may be the older age of patients with NCDs. This hypothesis is further supported by the fact that patients with cataracts or arthritis were older and generally less likely to practice satisfactory oral health behaviour and utilise oral health care services than other patients, whereas patients with allergic rhinitis or atopic dermatitis were younger and exhibited relatively higher rates of interdental care. However, this phenomenon is not necessarily independently attributable to age, because patients with dyslipidaemia, who were an older group, had better utilisation rates of annual scaling and oral examinations than those in the general population.

It should be noted that there are more vulnerable disease groups among various types of NCDs. For example, in 2013, the usage rate of dental floss and interdental brushes in stroke patients was only about 1/3 that of allergic rhinitis patients. In addition, in the case of utilisation of annual scaling or oral examinations experience, it was confirmed that the figures for cataract and arthritis patients were particularly low as of 2017. Although oral health care is known to be important for improving the health-related quality of life of stroke patients [30], it is expected that oral health care was not performed due to the nature of the disease, which is not easy to self-manage. Furthermore, in the case of arthritis patients, if there is a limitation of movement, it will not be easy to visit the dentist for oral care, and if there is a loss of functional ability of the hand, it will be difficult to maintain oral health behaviour [31].

## Policy suggestions

In Korea, education and promotion of oral health care among patients with NCDs are inadequate. Programmes encompassing the necessity of and methodology for appropriate oral health care among patients with NCDs are rarely provided at public health centres, oral health care services, hospitals, and clinics. Generally, hospitalisation for the management of a disease is an excellent opportunity to improve health behaviour, as a 'wake-up call' or 'teachable moment' [32]. Therefore, incorporating oral health care services into in- and outpatient management programmes for patients with NCDs will likely improve oral health care awareness.

As an example, a pilot chronic disease management programme targeting patients with hypertension and diabetes mellitus in primary care settings may be implemented by providing oral health management services in public health centres and dental clinics [33,34]. Such a plan can be expanded to the concept of medical-dental integration or oral medical care coordination [35]. In order to improve the health-related quality of life and mortality rate of NCDs patients, the establishment of an integrated prevention and management center to coordinate the provision of necessary health-medical-welfare services, including oral health care services, could be a practical alternative in Korea.

Furthermore, it is essential to implement measures to support vulnerable groups in terms of oral health. Among patients with NCDs, priority should be given to policy preparation targeting disease groups with worse oral health behaviour or lower rates of oral health service provision. This is especially true for disease groups in older age brackets. Additionally, for these more vulnerable disease groups, the level of oral health should be monitored more continuously, and oral education and services should be actively provided.

As revealed in this study, the substantial regional disparities in the practice of oral health behaviour and utilisation of oral health care services reveal the need to strengthen such behaviour and utilisation in specific, vulnerable areas. In 2017, in Seoul, a large city in Korea, the utilisation rate of annual scaling among patients with diabetes mellitus was 53.8%. However, in the rural and fishing province of Jeollanam-do, this rate was only 30.8%. These regional disparities may lead to differences in oral health levels in rural and urban areas [36]. Thus, in rural areas with limited resources, the efficacy of health-related service provision can be augmented by incorporating effective oral health care services into existing health-improvement programmes, thereby reducing regional disparities in oral health care [37]. Furthermore, it is necessary to monitor the oral health behaviour practice and oral health care service utilisation rate of NCD patients in rural areas more frequently to confirm whether the gap between regions is narrowing and to revise policies.

## Study limitations and scope for future research

This study had several limitations. First, the KCHS data analysed in this study included the presence of NCDs and health behaviours, which were self-reported. No additional procedures were performed to verify the validity of the data. Particularly, in this study, the rate of appropriate oral health behaviour might have been overestimated owing to the social desirability bias involved in self-reporting. Therefore, future qualitative research is required to verify the validity of self-reported oral health behaviour for patients with NCDs [38]. Furthermore, self-reporting can also be problematic when estimating the prevalence of NCDs. There is controversy over whether self-reporting can accurately estimate disease prevalence [39],

but it should be kept in mind that vulnerable patient groups that are not easily self-reported may be missed in the prevalence group.

Second, a causal relationship between the prevalence of NCDs and oral health behaviour could not be demonstrated in this study. We do not know the nature of the patients' oral health behaviour and oral health care service utilisation prior to their NCD diagnosis, or the subsequent changes in these items. For instance, patients might have had poor oral health behaviour before the diagnosis, which slightly improved after the onset of treatment for their NCD. Therefore, future studies, in which panel or patient registration data are used to compare oral health behaviour practices and oral health care service utilisation before and after NCD diagnosis, are necessary.

Third, as the KCHS does not regularly measure variables correlated with NCDs, oral health behaviour practices, and the utilisation of oral health services, trends cannot be analysed at the same intervals. Despite the importance of oral health, oral health-related responses were not regularly input into the KCHS database. Oral health-related questions should be asked on a more regular or annual basis to establish valid oral health care policies. Once sufficient data sources are available, it will be possible to conduct subgroup analyses, including cluster analysis, to identify more vulnerable groups among NCDs and the provision rate of oral health-related service can be identified to seek improvements in such services.

## Conclusion

Practising appropriate oral health behaviour and making adequate use of oral health care services may prevent some of the worsening of NCDs in patients. According to this study, among patients with NCDs, the practice and utilisation rates of oral health behaviour and oral health care services, respectively, are increasing. However, compared with those of the general population, further improvements are necessary. Institutionalising the promotion and education of oral health behaviour and oral health care service utilisation, and especially targeting patients with NCDs, may yield the desired improvements. There is a need to establish a system for medical-dental integration and oral medical care coordination for NCDs patients. Moreover, the rates of oral health behaviours and oral health care service utilisation in the general population and among patients with NCDs should be consistently monitored, both to evaluate the effectiveness of implemented policies and to make recommendations for which areas need to be improved.

## Supporting information

**Supplementary File 1.  Survey contents about non-communicable diseases and oral health behaviour by year.**
(XLSX)

**Supplementary File 2.  Toothbrushing questions by year.**
(XLSX)

**Supplementary File 3.  Oral health behaviour by year and region: toothbrushing practice.**
(XLSX)

**Supplementary File 4.  Oral health behaviour by year and region: use of dental floss and interdental brushes.**
(XLSX)

**Supplementary File 5.  Oral health service utilisation by year and region: annual scaling and oral examinations.**
(XLSX)

## Author contributions

**Conceptualization:** Jeehee Pyo, Hyeran Jeong, Noor Afif Mahmudah, Young-Kwon Park, Minsu Ock.

**Data curation:** Jeehee Pyo, Noor Afif Mahmudah, Young-Kwon Park, Minsu Ock.

**Formal analysis:** Young-Kwon Park, Minsu Ock.

**Funding acquisition:** Minsu Ock.

**Investigation:** Jeehee Pyo, Hyeran Jeong, Minsu Ock.

**Methodology:** Young-Kwon Park, Minsu Ock.

**Project administration:** Minsu Ock.

**Supervision:** Minsu Ock.

**Validation:** Jeehee Pyo, Hyeran Jeong, Minsu Ock.

**Visualization:** Jeehee Pyo, Hyeran Jeong, Minsu Ock.

**Writing – original draft:** Jeehee Pyo, Hyeran Jeong, Minsu Ock.

**Writing – review & editing:** Jeehee Pyo, Hyeran Jeong, Noor Afif Mahmudah, Young-Kwon Park, Minsu Ock.

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
