## [Decision Letter · Decision Letter 0]

8 Nov 2024

PONE-D-24-31397Comparative analysis of oral health behaviour and utilisation of oral health care services in the general population and among patients with non-communicable diseases in Korea: a repeated cross-sectional survey conducted from 2008 to 2022PLOS ONE

Dear Dr. Ock,

Thank you for submitting your manuscript to PLOS ONE. After careful consideration, we feel that it has merit but does not fully meet PLOS ONE’s publication criteria as it currently stands. Therefore, we invite you to submit a revised version of the manuscript that addresses the points raised during the review process.

We look forward to receiving your revised manuscript.

Kind regards,

Pracheth Raghuveer, MD, DNB

Academic Editor

PLOS ONE

**Journal Requirements:**

This study was supported by a grant from the National R&D Program for Cancer Control, Ministry of Health & Welfare, Republic of Korea (HA21C0107). The funders had no role in study design, data collection and analysis, decision to publish, or preparation of the manuscript.

Reviewers' comments:

Reviewer's Responses to Questions

**Comments to the Author**

1. Is the manuscript technically sound, and do the data support the conclusions?

Reviewer #1: Yes

Reviewer #2: Yes

2. Has the statistical analysis been performed appropriately and rigorously? 

Reviewer #1: Yes

Reviewer #2: Yes

3. Have the authors made all data underlying the findings in their manuscript fully available?

Reviewer #1: Yes

Reviewer #2: Yes

4. Is the manuscript presented in an intelligible fashion and written in standard English?

Reviewer #1: Yes

Reviewer #2: Yes

5. Review Comments to the Author

**Reviewer #1: ** The study provides a crucial and timely examination of oral health behavior and service utilization among patients with non-communicable diseases (NCDs). Given the global rise in NCDs, this research holds significant relevance to both clinical practice and public health, highlighting the disparities in oral health care that may exist for this vulnerable group. The analysis is bolstered by the use of a robust dataset from the Korea Community Health Survey spanning a substantial 14-year period (2008–2022), offering a comprehensive view of trends over time. This extended period enables meaningful insights into how oral health behaviors and utilization of services have evolved, both in the general population and in patients with NCDs.

One of the study's major strengths is its identification of the disparities in oral health behaviors between the general population and those with NCDs. By demonstrating that NCD patients generally exhibit poorer oral health behaviors, the research points to specific areas for intervention, which is essential for developing tailored strategies to improve their oral health outcomes. Furthermore, the study has significant public health implications, as it provides an evidence-based foundation for healthcare policymakers. Identifying gaps in oral health services for NCD patients paves the way for strategies that can enhance preventive care and improve access to services, ultimately leading to better overall health outcomes for these patients.

However, some critical points require further attention to maximize the study's impact:

Lack of Specificity in Patient Subgroups: While the analysis covers 15 different NCDs, the abstract does not clarify whether specific trends or significant differences were observed among the various NCD subgroups. It would strengthen the study to explore whether particular conditions, such as diabetes mellitus or hypertension, have distinct oral health behaviors or service utilization patterns. Additionally, these 15 NCDs could be grouped into major categories (such as metabolic, cardiovascular, and mental health conditions) for more granular analysis. Since oral diseases share common risk factors with several NCDs, a subgroup analysis could reveal important clusters of risk factors and provide insights into how interventions might be more effectively targeted.

Generalized Recommendations: The abstract concludes with a recommendation to implement educational and promotional measures, but these suggestions are somewhat broad. Providing more specific or innovative strategies tailored to the needs of NCD patients could enhance the recommendations' impact. For instance, highlighting which specific oral health behaviors or service utilization gaps—such as annual scaling or regular oral examinations—should be prioritized for intervention would make the recommendations more actionable.

Comparison with General Population: Although the comparison between NCD patients and the general population is briefly mentioned, the abstract does not delve deeply into the implications of these differences. A more thorough discussion of why oral health service utilization rates are lower in NCD patients, whether due to access issues, awareness gaps, or other barriers, could provide valuable insights that inform future public health strategies.

Limited Focus on Policy Implications: While the study emphasizes the need for improved oral health behaviors in NCD patients, it could better emphasize the potential policy changes or adjustments in healthcare systems necessary to address these gaps. Given the increasing global burden of NCDs, incorporating a discussion on the importance of medical-dental integration and coordinated care for patients with NCDs would significantly strengthen the study. Policy suggestions that focus on better integrating dental care within overall NCD management could offer long-term solutions for improving health outcomes in this population.

The following suggestions may further enhance the clarity and impact of the study:

Provide more specificity on differences in oral health behavior and service utilization across distinct NCD subgroups to offer more targeted insights. (to explore whether Major NCDs and other NCDs can be subgroup in the analysis )

Strengthen the policy implications by discussing potential strategies for medical-dental integration and more coordinated care for NCD patients.

Clarify and expand upon the regional variations identified, including potential reasons for these disparities and tailored approaches to address them.

These revisions will further solidify the paper’s contribution to the field and its practical relevance for addressing oral health gaps in NCD care.

**Reviewer #2: ** 1. The authors have used available survey data from community health surveys and tried to capture the trends/changes in oral health behaviour among those with non communicable diseases (NCDs) and among the other general population. A problem with public health surveys as compared to registry-like data is that the sample is bound to change every year, (900 who are sampled per public health center, will change every year!). Within the limits of the available data, the authors have tried to capture the trends using a suitable statistical trend analysis/regression.

2. The main issue in the analysis that I noticed is that the change in oral health behaviours is on the up-trend for all population but, those with NCDs it is still less. In addition to the "self reported" nature and the associated false reporting by people, the other thing is that those with highest risk (among the NCD and non NCD population) may not be captured because they are either not coming into the system or they are lost... Also, this self-reporting will also under-report those who have not yet been identified or diagnosed with NCD, which in high probabiility are people who do not access the health system. These limitations should reflect in the discussion.

3. Some NCDs have a higher morbidity and higher association with poor oral health. In addition, NCDs being a lifestyle related disease, especially hypertension, diabetes (DM); those with common risk factors may also have a poor oral health behaviour and this may reflect on their overall low use of health resources and their consequent poor oral health. The change in behaviour would not happen without targetted interventions. And hence, they will remain low. Some segments of population with mental health challenges, may also resort to poor self-care and ignore their usual care routines, including tooth brushing etc., The authors could explain such associations in the discussion. They have explained the possible causes for poor oral health habits among those with stroke, but different NCDs may affect their ability to maintain or access the system in different ways. For example, arthritis may affect their dexterity to brush well and also their mobility and hence decreased access. Those with another NCDs may have different challenge, and is not discussed even in broad sense. A table to highlight this or a short paragraph on the same will be good, and this should be loco-regionally informed by prior data or known experience of the authors and not just speculation.

4. I feel, the policy level suggestions and system level changes suggested by the authors based on their work appear too generic and not data driven- as per the data presented. Also, with so much missing data in so many years and with information only on 'tooth brushing habits', it is difficult to justify major policy level recommendations. The strongest point is the recommendation that oral health has to be integrated with NCD screening. Majority of individuals in general population and those with NCDs have shown similar habits except some specific conditions such as stroke. The authors have themselves identified potential issues in this group of population. They have found a reverse trend in patients with arthritis & dyspidemia. The potential reason is that arthritis & dyslipidemia have both increased in prevalence since 2008, and a large number of individuals with good oral health habits may also have developed dyslipidemia and are now here, where-in their habits has not changed, just their categorisation. With respect to arthritis, the reason for increased in oral examination in previous years may be due to the pre-joint replacement oral health screening, to remove foci of infections prior to reconstruction/replacement (or possibly some other reasons).

The authors have included 15 NCDs and have tried to explain a lot of changes, but during the discussion and conclusion they resort to generic suggestions than to base it on the data along with known trends in NCDs in their country. A suggestion would be to correlate the oral health behaviours to changing prevalence in NCDs which may reveal whether those with NCDs are actually neglecting their oral health. It is also prudent to highlight those conditions where the impact of poor oral health can be more detrimental. Atopic dermatitis, allergic rhinitis conditions with a poor oral health behaviour is a problem, but not so much as a person with DM or hypertension or stroke, where the risks may be higher.

The overall impact of the paper and the policy level suggestions they can provide can be better improved if the data is interprepted or discussed based on prevalence and change in prevalence of the NCDs themselves especially with respect to those with higher morbidity and stronger established association with poor oral health such as DM.

5. What regional factors/system level factors affect their poor utilisation ? Is there poor access ? Or some other reasons, this is best explained by authors since they know the system better at their country.

6. PLOS authors have the option to publish the peer review history of their article (what does this mean? ). If published, this will include your full peer review and any attached files.

**Do you want your identity to be public for this peer review?** For information about this choice, including consent withdrawal, please see our Privacy Policy .

Reviewer #1: No

Reviewer #2: **Yes: ** Akilesh. R

---

## [Author Response · Author response to Decision Letter 1]

24 Dec 2024

Review Comments to the Author

Response: We would like to express our sincere gratitude for your valuable feedback and constructive comments on our manuscript. We have revised our manuscript in accordance with reviewers’ suggestions.

Reviewer #1: The study provides a crucial and timely examination of oral health behavior and service utilization among patients with non-communicable diseases (NCDs). Given the global rise in NCDs, this research holds significant relevance to both clinical practice and public health, highlighting the disparities in oral health care that may exist for this vulnerable group. The analysis is bolstered by the use of a robust dataset from the Korea Community Health Survey spanning a substantial 14-year period (2008–2022), offering a comprehensive view of trends over time. This extended period enables meaningful insights into how oral health behaviors and utilization of services have evolved, both in the general population and in patients with NCDs.

One of the study's major strengths is its identification of the disparities in oral health behaviors between the general population and those with NCDs. By demonstrating that NCD patients generally exhibit poorer oral health behaviors, the research points to specific areas for intervention, which is essential for developing tailored strategies to improve their oral health outcomes. Furthermore, the study has significant public health implications, as it provides an evidence-based foundation for healthcare policymakers. Identifying gaps in oral health services for NCD patients paves the way for strategies that can enhance preventive care and improve access to services, ultimately leading to better overall health outcomes for these patients.

Response: Thank you for your valuable feedback. As mentioned, we expect that this manuscript will contribute to establishing strategies to improve oral health in NCD patients. The manuscript has been revised according to your comments below.

However, some critical points require further attention to maximize the study's impact:

Lack of Specificity in Patient Subgroups: While the analysis covers 15 different NCDs, the abstract does not clarify whether specific trends or significant differences were observed among the various NCD subgroups. It would strengthen the study to explore whether particular conditions, such as diabetes mellitus or hypertension, have distinct oral health behaviors or service utilization patterns. Additionally, these 15 NCDs could be grouped into major categories (such as metabolic, cardiovascular, and mental health conditions) for more granular analysis. Since oral diseases share common risk factors with several NCDs, a subgroup analysis could reveal important clusters of risk factors and provide insights into how interventions might be more effectively targeted.

Response: Thank you for your valuable suggestion. As you mentioned, it is important to examine which types of NCD patients have more problems with oral health behaviour. Therefore, we focused more on the oral health behaviour of patients with hypertension or diabetes, which are the most common types of NCD patients (Page 14~). However, we acknowledge that the current Discussion section is categorized based on individual oral health behaviour rather than disease-centered. Accordingly, we added more content to Discussion section to monitor the oral health behaviour of more vulnerable disease groups among various types of NCD patients (Page 21).

Furthermore, it would be meaningful to divide the types of NCD patients and check whether there are differences in oral health behaviour between subgroups. However, the primary purpose of this study was to focus on comparing and analyzing trends in health behaviour in various types of NCD patients. Therefore, we did not perform statistical analyses such as cluster analysis to divide the types of NCD patients separately. However, we believe that your suggestions are worth considering in future studies. We have added suggestions for subgroup analysis in Discussion section (Page 24).

Generalized Recommendations: The abstract concludes with a recommendation to implement educational and promotional measures, but these suggestions are somewhat broad. Providing more specific or innovative strategies tailored to the needs of NCD patients could enhance the recommendations' impact. For instance, highlighting which specific oral health behaviors or service utilization gaps—such as annual scaling or regular oral examinations—should be prioritized for intervention would make the recommendations more actionable.

Response: As you suggested, we have made our suggestions in Conclusion section of the Abstract more specific (Page 5).

Comparison with General Population: Although the comparison between NCD patients and the general population is briefly mentioned, the abstract does not delve deeply into the implications of these differences. A more thorough discussion of why oral health service utilization rates are lower in NCD patients, whether due to access issues, awareness gaps, or other barriers, could provide valuable insights that inform future public health strategies.

Response: The abstract, which has a word count limit, could not fully cover the results. We checked whether the results section of the abstract was more specific and revised the description accordingly (Page 4).

Limited Focus on Policy Implications: While the study emphasizes the need for improved oral health behaviors in NCD patients, it could better emphasize the potential policy changes or adjustments in healthcare systems necessary to address these gaps. Given the increasing global burden of NCDs, incorporating a discussion on the importance of medical-dental integration and coordinated care for patients with NCDs would significantly strengthen the study. Policy suggestions that focus on better integrating dental care within overall NCD management could offer long-term solutions for improving health outcomes in this population.

Response: We also strongly agree with the importance of medical-dental integration and coordinated care for patients with NCDs that you mentioned. That is why we have already emphasized the importance of oral health care services in in- and outpatient management programmes for patients with NCDs (Page 22). Here, we have put more emphasis on the integrated provision of oral health care services.

The following suggestions may further enhance the clarity and impact of the study:

Provide more specificity on differences in oral health behavior and service utilization across distinct NCD subgroups to offer more targeted insights. (to explore whether Major NCDs and other NCDs can be subgroup in the analysis)

Strengthen the policy implications by discussing potential strategies for medical-dental integration and more coordinated care for NCD patients.

Clarify and expand upon the regional variations identified, including potential reasons for these disparities and tailored approaches to address them.

These revisions will further solidify the paper’s contribution to the field and its practical relevance for addressing oral health gaps in NCD care.

Response: Thank you again for your progressive suggestions that can enhance the policy application of the research. In addition to the subgroup analysis of NCDs that you suggested earlier, we believe that medical-dental integration and more coordinated care for NCD patients are review comments that can develop the results of this study into actual policies. We reviewed and revised the related descriptions in Discussion section to include specific policy suggestions (Page 22).

Furthermore, we agree with the opinion that we should pay more attention to the differences in oral health behaviour among NCD patients between regions confirmed in this study. Therefore, we separately described the differences in oral health behaviour among provinces and cities in Korea in Results section, and emphasized the need to prepare measures for vulnerable areas of oral health behaviour and oral health care services in Discussion section (Page 22). We reexamined the description of these contents and refined them to deliver a clearer message (Page 23)

Reviewer #2: 1. The authors have used available survey data from community health surveys and tried to capture the trends/changes in oral health behaviour among those with non communicable diseases (NCDs) and among the other general population. A problem with public health surveys as compared to registry-like data is that the sample is bound to change every year, (900 who are sampled per public health center, will change every year!). Within the limits of the available data, the authors have tried to capture the trends using a suitable statistical trend analysis/regression.

Response: Thank you for your valuable feedback. As you pointed out, the Korea Community Health Survey does not use a panel survey method but rather a repetitive cross-sectional survey method. We have already mentioned in Discussion section that this survey method has a limitation in that it is difficult to identify the causal relationship between disease diagnosis and oral health behavior (Page 23). However, the large sample size of the Korea Community Health Survey can be seen as increasing the validity and generalizability of the analysis results. This methodological advantage has also been mentioned in Discussion section (Page 18).

2. The main issue in the analysis that I noticed is that the change in oral health behaviours is on the up-trend for all population but, those with NCDs it is still less. In addition to the "self reported" nature and the associated false reporting by people, the other thing is that those with highest risk (among the NCD and non NCD population) may not be captured because they are either not coming into the system or they are lost... Also, this self-reporting will also under-report those who have not yet been identified or diagnosed with NCD, which in high probabiility are people who do not access the health system. These limitations should reflect in the discussion.

Response: The Korea Community Health Survey relies on self-reports to collect responses related to health behaviors. This limitation may result in low validity of the responses. We have already addressed this as a possibility of social desirability bias in Discussion section (Page 23). However, as you mentioned, self-reports may also affect the determination of whether or not a person has a disease. In other words, we need to be careful when estimating the prevalence of NCD patients. We have added this limitation to Discussion section (Page 23).

3. Some NCDs have a higher morbidity and higher association with poor oral health. In addition, NCDs being a lifestyle related disease, especially hypertension, diabetes (DM); those with common risk factors may also have a poor oral health behaviour and this may reflect on their overall low use of health resources and their consequent poor oral health. The change in behaviour would not happen without targetted interventions. And hence, they will remain low. Some segments of population with mental health challenges, may also resort to poor self-care and ignore their usual care routines, including tooth brushing etc., The authors could explain such associations in the discussion. They have explained the possible causes for poor oral health habits among those with stroke, but different NCDs may affect their ability to maintain or access the system in different ways. For example, arthritis may affect their dexterity to brush well and also their mobility and hence decreased access. Those with another NCDs may have different challenge, and is not discussed even in broad sense. A table to highlight this or a short paragraph on the same will be good, and this should be loco-regionally informed by prior data or known experience of the authors and not just speculation.

Response: In this study, we examined the trends in oral health behaviours and oral health care service utilisation for patients with various types of NCDs. We also summarized the implications for each oral health behavior and oral health care service utilisation. Although we presented the results focusing on diabetes and hypertension, which have large patient volumes among NCDs, we acknowledge that we were unable to examine the characteristics of each disease in detail. Accordingly, we added more content to Discussion section to monitor the oral health behaviour of more vulnerable disease groups among various types of NCDs patients (Page 21).

4. I feel, the policy level suggestions and system level changes suggested by the authors based on their work appear too generic and not data driven- as per the data presented. Also, with so much missing data in so many years and with information only on 'tooth brushing habits', it is difficult to justify major policy level recommendations. The strongest point is the recommendation that oral health has to be integrated with NCD screening. Majority of individuals in general population and those with NCDs have shown similar habits except some specific conditions such as stroke. The authors have themselves identified potential issues in this group of population. They have found a reverse trend in patients with arthritis & dyspidemia. The potential reason is that arthritis & dyslipidemia have both increased in prevalence since 2008, and a large number of individuals with good oral health habits may also have developed dyslipidemia and are now here, where-in their habits has not changed, just their categorisation. With respect to arthritis, the reason for increased in oral examination in previous years may be due to the pre-joint replacement oral health screening, to remove foci of infections prior to reconstruction/replacement (or possibly some other reasons).

Response: We have presented the policy application of this study in a separate part called Policy Suggestions in Discussion section. As you suggested, there is a way to simultaneously perform NCDs screening and oral examination, but this study focused on the value of hospitalization and outpatient visits due to treatment of NCDs patients (Page 22). In a previous study, this was called a ‘wake-up call’ or ‘teachable moment’ and suggested as a good opportunity to improve the patients’ health behaviours. Therefore, we mentioned that the importance of oral health behaviours and the necessity of using oral health care services should be sufficiently addressed in the education of NCD patients in the inpatient and outpatient settings. However, we have revised and strengthened the policy suggestions considering your suggestion (Page 22).

The authors have included 15 NCDs and have tried to explain a lot of changes, but during the discussion and conclusion they resort to generic suggestions than to base it on the data along with known trends in NCDs in their country. A suggestion would be to correlate the oral health behaviours to changing prevalence in NCDs which may reveal whether those with NCDs are actually neglecting their oral health. It is also prudent to highlight those conditions where the impact of poor oral health can be more detrimental. Atopic dermatitis, allergic rhinitis conditions with a poor oral health behaviour is a problem, but not so much as a person with DM or hypertension or stroke, where the risks may be higher.

The overall impact of the paper and the policy level suggestions they can provide can be better improved if the data is interprepted or discussed based on prevalence and change in prevalence of the NCDs themselves especially with respect to those with higher morbidity and stronger established association with poor oral health such as DM.

Response: As you mentioned, changes in oral health behaviours and oral health care service utilisation should be considered together with changes in the prevalence of NCDs. However, as mentioned earlier, the Korea Community Health Survey, the data source f

---

## [Decision Letter · Decision Letter 1]

10 Jan 2025

PONE-D-24-31397R1Comparative analysis of oral health behaviour and utilisation of oral health care services in the general population and among patients with non-communicable diseases in Korea: a repeated cross-sectional survey conducted from 2008 to 2022PLOS ONE

Dear Dr. Ock,

Thank you for submitting your manuscript to PLOS ONE. After careful consideration, we feel that it has merit but does not fully meet PLOS ONE’s publication criteria as it currently stands. Therefore, we invite you to submit a revised version of the manuscript that addresses the points raised during the review process.

We look forward to receiving your revised manuscript.

Kind regards,

Pracheth Raghuveer, MD, DNB

Academic Editor

PLOS ONE

Journal Requirements:

Reviewers' comments:

Reviewer's Responses to Questions

**Comments to the Author**

1. If the authors have adequately addressed your comments raised in a previous round of review and you feel that this manuscript is now acceptable for publication, you may indicate that here to bypass the “Comments to the Author” section, enter your conflict of interest statement in the “Confidential to Editor” section, and submit your "Accept" recommendation.

Reviewer #1: All comments have been addressed

Reviewer #2: All comments have been addressed

2. Is the manuscript technically sound, and do the data support the conclusions?

Reviewer #1: Yes

Reviewer #2: Yes

3. Has the statistical analysis been performed appropriately and rigorously? 

Reviewer #1: Yes

Reviewer #2: Yes

4. Have the authors made all data underlying the findings in their manuscript fully available?

Reviewer #1: Yes

Reviewer #2: Yes

5. Is the manuscript presented in an intelligible fashion and written in standard English?

Reviewer #1: Yes

Reviewer #2: Yes

6. Review Comments to the Author

Reviewer #1: the comments to reviewer is satisfactory and the article presents important findings for medical dental integration.

Reviewer #2: The authors have addressed the major concerns adequately. A few additional points for revision are submitted for the consideration of the authors.

1. Abstract - Result section can be improved and may consider the following suggestions.

First part of the result section is on oral health behaviour. So, logically, the second can start with "Regarding Oral health service utilisation..." and follow it up with key results. The third part can be on "inter-regional" variations.

2. Results pg 11 Line 4: From the study design, the total number is NOT 3,428, 968 patients, but datapoints (since around 200,000 per year are surveyed, with substantial overlap every year, which would be difficult to track). Unless, the survey method avoids resampling the same individual. And may consider either rephrasing the line 4 for total number of patients or may remove that from results to avoid confusion. Or may report the number of individuals sampled in the last survey alone. Or can mention, number surveyed in 2008 and the number surveyed in 2022.

Similar issues with the numbers presented for HTN, DM, arthritis etc., I think, the authors can use the LATEST survey result to summarise this data for the Table 1 and for reporting purposes. This gives the current or recent information that may be useful for policy decisions. The number of individuals with specific conditions in 2008 is less relevant compared to the number of people with the condition/disease in 2022. This revision will make the numbers easier to interpret. Table 1 may be updated in similar manner with latest data only, I feel.

4. Oral Health Behavior results: pg 14-15

The trends can be explained as a summary and since figures are self-explanatory, the important data alone may be highlighted and rest mentioned or labeled in figures. The information that is removed during this editing may be added to a supplement file under same headings.

Oral Health service utilisation and Inter-regional indicators may remain with sufficient detail as it is in the main article.

5. Discussion: pg 18 Line 3: Main findings & significance of the study - The heading is redundant.

6. pg 18 Line 4-6: The first sentence of discussion shall be the result and not what the authors did and may be removed or moved to next paragraph. The discussion may begin with line 7 in this page where the authors are summarising the study findings.

7. The limitations section may be brought up before "interpretation and implications" so the reader may understand the limitations before reading the interpretations, and then policy suggestions followed by conclusion. Necessary minor changes may be needed to maintain the logical flow of the article/writing.

8. pg 19-21 Implications This section has repeated much of the "Result section" information including the numbers. This repetition is redundant. This section can remove most of the statistical data, except the ones that are pivotal for further "policy recommendations" or "informing future research". Further sections on implications may also follow similar strategies of summary and implication instead of repeating the data from results section again.

Very important, very large or unexpected results/data may be retained for contextual purposes such as three fold increase or alarming decrease to emphasize or compare some key data. Otherwise, repetition of results data is redundant in discussion. In most paragraphs under "Implications", the second half of each paragraph contains information relevant to discussion section and first half has repetition from Results section. These lines may be removed for brevity and to remove redundancy.

9. Pg 21; lines 6-9, 19,20 May be shifted to policy suggestion section and may add about "improving access".

10. pg 21-23 Policy Suggestions

The policy recommendation section may be shifted to just before conclusion. Vulnerable and marginalized population specific recommendations may mention about improving access and promoting preventive oral care.

11. The third limitation from pg 24 can inform policy recommendation about integrating medical-dental care and about need to include dental/oral health related items in the KCHS surveys. Bridging this data-gap is an important policy recommendation, I feel.

12. pg 25 Lines 1-13 Conclusion: I feel that the oral Health behaviour and utilising oral health services alone are not sufficient to prevent worsening of NCDs and I feel that the data in the study doesn't support the first line of conclusion. The authors may instead add a note on "common risk factors" as a link between oral health and general health. May take this suggestion, if agreeable, since this may appeal to a larger audience and it will be an effective way to push for integration of dental and medical care in the health system. Following this, the authors can also add a note on "integration of medical-dental health" in conclusion. In addition, the repetitive sentences from other sections of discussion may be removed or summarised and rephrased briefly. Lines 8-13 appear good and may be retained.

7. PLOS authors have the option to publish the peer review history of their article (what does this mean? ). If published, this will include your full peer review and any attached files.

**Do you want your identity to be public for this peer review?** For information about this choice, including consent withdrawal, please see our Privacy Policy .

Reviewer #1: No

Reviewer #2: **Yes: ** Akilesh R

---

## [Author Response · Author response to Decision Letter 2]

18 Feb 2025

Reviewer #1: the comments to reviewer is satisfactory and the article presents important findings for medical dental integration.

Response: We sincerely appreciate your time and effort in reviewing our manuscript. Thank you once again for your careful evaluation and valuable input.

Reviewer #2: The authors have addressed the major concerns adequately. A few additional points for revision are submitted for the consideration of the authors.

Response: We are grateful for your constructive feedback and suggestions. The manuscript has been revised according to your comments below.

1. Abstract - Result section can be improved and may consider the following suggestions.

First part of the result section is on oral health behaviour. So, logically, the second can start with "Regarding Oral health service utilisation..." and follow it up with key results. The third part can be on "inter-regional" variations.

Response: As you suggested, we have revised the sentences in the Abstract (page 4).

2. Results pg 11 Line 4: From the study design, the total number is NOT 3,428, 968 patients, but datapoints (since around 200,000 per year are surveyed, with substantial overlap every year, which would be difficult to track). Unless, the survey method avoids resampling the same individual. And may consider either rephrasing the line 4 for total number of patients or may remove that from results to avoid confusion. Or may report the number of individuals sampled in the last survey alone. Or can mention, number surveyed in 2008 and the number surveyed in 2022.

Similar issues with the numbers presented for HTN, DM, arthritis etc., I think, the authors can use the LATEST survey result to summarise this data for the Table 1 and for reporting purposes. This gives the current or recent information that may be useful for policy decisions. The number of individuals with specific conditions in 2008 is less relevant compared to the number of people with the condition/disease in 2022. This revision will make the numbers easier to interpret. Table 1 may be updated in similar manner with latest data only, I feel.

Response: We agree with the problem with the presentation method of Table 1. Instead of the existing method of presenting the number of patients by age group by disease, we have revised Table 1 to present the number of patients by year by disease. Accordingly, we have revised the sentences in the Results to emphasize the number of patients in the most recent year, as you suggested (page 11-15).

4. Oral Health Behavior results: pg 14-15

The trends can be explained as a summary and since figures are self-explanatory, the important data alone may be highlighted and rest mentioned or labeled in figures. The information that is removed during this editing may be added to a supplement file under same headings.

Oral Health service utilisation and Inter-regional indicators may remain with sufficient detail as it is in the main article.

Response: Taking your feedback into account, we have revised the sentence to make it more concise by removing duplicate numbers (page 16).

5. Discussion: pg 18 Line 3: Main findings & significance of the study - The heading is redundant.

Response: As suggested, we have removed that subheadings (page 20).

6. pg 18 Line 4-6: The first sentence of discussion shall be the result and not what the authors did and may be removed or moved to next paragraph. The discussion may begin with line 7 in this page where the authors are summarising the study findings.

Response: We have removed the first sentence in the Discussion section based on your review comment (page 20).

7. The limitations section may be brought up before "interpretation and implications" so the reader may understand the limitations before reading the interpretations, and then policy suggestions followed by conclusion. Necessary minor changes may be needed to maintain the logical flow of the article/writing.

Response: Thank you for your feedback. Depending on your preference, we would be happy to present the “Study limitations and scope for future research” before the “Interpretation and implications of principal findings.” However, it is customary to present the study limitations and suggestions for future research at the end of the Discussion section. We kindly ask for your understanding.

8. pg 19-21 Implications This section has repeated much of the "Result section" information including the numbers. This repetition is redundant. This section can remove most of the statistical data, except the ones that are pivotal for further "policy recommendations" or "informing future research". Further sections on implications may also follow similar strategies of summary and implication instead of repeating the data from results section again.

Very important, very large or unexpected results/data may be retained for contextual purposes such as three fold increase or alarming decrease to emphasize or compare some key data. Otherwise, repetition of results data is redundant in discussion. In most paragraphs under "Implications", the second half of each paragraph contains information relevant to discussion section and first half has repetition from Results section. These lines may be removed for brevity and to remove redundancy.

Response: We sincerely thank you for your valuable feedback. In response, we have reviewed and revised the sentences in the “Interpretation and implications of principal findings” section of the Discussion. Specifically, we have made the content more concise by removing unnecessary repetitions of the results (page 21).

9. Pg 21; lines 6-9, 19,20 May be shifted to policy suggestion section and may add about "improving access".

Response: In response to your feedback, we have relocated this content to “Policy suggestions” section (page 23-24).

10. pg 21-23 Policy Suggestions

The policy recommendation section may be shifted to just before conclusion. Vulnerable and marginalized population specific recommendations may mention about improving access and promoting preventive oral care.

Response: As mentioned earlier, we kindly ask for your understanding that the “Study limitations and scope for future research” is placed at the end of the Discussion section. We appreciate your understanding once again, as we have addressed both the limitations of the study and suggestions for future research simultaneously.

11. The third limitation from pg 24 can inform policy recommendation about integrating medical-dental care and about need to include dental/oral health related items in the KCHS surveys. Bridging this data-gap is an important policy recommendation, I feel.

Response: Thank you for your valuable suggestion. We have already emphasized medical-dental integration and oral medical care coordination in the first revision. As per your recommendation, we agree that it is crucial to conduct regular oral health-related surveys to monitor the oral health services that NCD patients are receiving, or not receiving. To reflect this, we have added a sentence highlighting the importance of monitoring the provision rate of oral health-related services (page 25).

12. pg 25 Lines 1-13 Conclusion: I feel that the oral Health behaviour and utilising oral health services alone are not sufficient to prevent worsening of NCDs and I feel that the data in the study doesn't support the first line of conclusion. The authors may instead add a note on "common risk factors" as a link between oral health and general health. May take this suggestion, if agreeable, since this may appeal to a larger audience and it will be an effective way to push for integration of dental and medical care in the health system. Following this, the authors can also add a note on "integration of medical-dental health" in conclusion. In addition, the repetitive sentences from other sections of discussion may be removed or summarised and rephrased briefly. Lines 8-13 appear good and may be retained.

Response: We sincerely appreciate your feedback. Taking it into account, we have revised the sentences in the Conclusion section (page 26). In particular, we have placed greater emphasis on the importance of medical-dental integration and oral medical care coordination for patients with NCDs.

---

## [Editor Report · Decision Letter 2]

12 Mar 2025

Comparative analysis of oral health behaviour and utilisation of oral health care services in the general population and among patients with non-communicable diseases in Korea: a repeated cross-sectional survey conducted from 2008 to 2022

PONE-D-24-31397R2

Dear Dr. Ock,

We’re pleased to inform you that your manuscript has been judged scientifically suitable for publication and will be formally accepted for publication once it meets all outstanding technical requirements.

Kind regards,

Pracheth Raghuveer, MD, DNB

Academic Editor

PLOS ONE
---

## [Editor Report · Acceptance letter]

PONE-D-24-31397R2

PLOS ONE

Dear Dr. Ock,

I'm pleased to inform you that your manuscript has been deemed suitable for publication in PLOS ONE. Congratulations! Your manuscript is now being handed over to our production team.

Kind regards,

on behalf of

Dr. Pracheth Raghuveer

Academic Editor

PLOS ONE
